# Dozens of Translation Directions or Millions of Shared Parameters? Comparing Two Types of Multilinguality in Modular Machine Translation

**Michele Boggia**♠ **Stig-Arne Grönroos**♠ **Niki Andreas Loppi**◇ **Timothee Mickus**♠

**Alessandro Raganato**♡ **Jörg Tiedemann**♠ **Raúl Vázquez**♠

♠`{firstname.lastname}@helsinki.fi` ◇`nloppi@nvidia.com`
♡`alessandro.raganato@unimib.it`

## Abstract

There are several ways of implementing multilingual NLP systems but little consensus as to whether different approaches exhibit similar effects. Are the trends that we observe when adding more languages the same as those we observe when sharing more parameters? We focus on encoder representations drawn from modular multilingual machine translation systems in an English-centric scenario, and study their quality from multiple aspects: how adequate they are for machine translation, how independent of the source language they are, and what semantic information they convey. Adding translation directions in English-centric scenarios does not conclusively lead to an increase in translation quality. Shared layers increase performance on zero-shot translation pairs and lead to more language-independent representations, but these improvements do not systematically align with more semantically accurate representations, from a monolingual standpoint.

## 1 Introduction

Multilinguality, within the scope of neural NLP, can mean either ensuring that computations for different languages are homogeneous, or ensuring that models are trained with data coming from different languages. These two definitions are not as equivalent as they might appear: for instance, modular architectures, where some parameters are specific to a single language, can only be conceived as multilingual under the latter definition.

Both of these trends have been explored across multiple works. Machine translation studies have looked into sharing no parameters at all (Escolano et al., 2021; Lyu et al., 2020), sharing across linguistically informed groups (Fan et al., 2021; Purason and Tättar, 2022), sharing only some components across all languages (Dong et al., 2015; Firat et al., 2016; Vázquez et al., 2020; Liao et al., 2021; Zhu et al., 2020; Kong et al., 2021; Blackwood et al., 2018; Sachan and Neubig, 2018; Zhang et al., 2021), and sharing the entire model (Johnson et al., 2017). Concerns about multilinguality have spearheaded research on how to make representations and systems more reliable for typologically and linguistically diverse data (Bojanowski et al., 2017; Adelani et al., 2022), the distinction between multilingual and monolingual representations (Wu and Dredze, 2020), the specificity of massively-multilingual representations (Kudugunta et al., 2019) or the effects of having more diverse data (Arivazhagan et al., 2019; Aharoni et al., 2019; Costa-jussà et al., 2022; Siddhant et al., 2022; Kim et al., 2021; Voita et al., 2019). In this paper, we study whether these different implementations of multilinguality yield qualitatively different types of representations—in other words: *Are the effects of parameter sharing orthogonal to those of adding new languages?*

To broach this question, we make three simplifying assumptions. *First*, we only consider the task of multilingual machine translation—an exhaustive study of the impact of all multilingual NLP tasks is beyond the scope of this paper. Moreover, massively multilingual language models are known to leverage parallel data to enhance semantic abstractions (Hu et al., 2021; Ouyang et al., 2021; Kale et al., 2021). *Second*, we only consider parameter sharing in the last layers of the encoders: we focus on the intermediary representations acquired directly after the encoder and leave decoders for future study. As language selection tokens would compromise the language independence of the representations, this rules out fully shared decoders.

---

Authors listed alphabetically. Corresponding author: `timothee.mickus@helsinki.fi`

*Third*, we focus on an English-centric scenario: i.e., all translation directions seen during training contain English as a source or target language. While such an approach is not without issues (Gu et al., 2019; Zhang et al., 2020), it makes it possible to select translation directions for zero-shot evaluations in a principled manner. Furthermore, most multilingual translation datasets are highly skewed in any case and contain orders of magnitude more English examples (e.g., Costa-jussà et al., 2022).

We conduct our study by testing encoder outputs on three aspects: *task fitness*, *language independence* and *semantic content*. These features have been discussed in earlier literature: probing pretrained language models for semantic content in particular has proven very fecund (e.g., Rogers et al., 2021; Doddapaneni et al., 2021). As for machine translation, these studies are less numerous, although similar aspects have been investigated (Raganato and Tiedemann, 2018). For instance, Kudugunta et al. (2019) study how the learned representations evolve in a multilingual scenario, whereas Vázquez et al. (2020), Raganato et al. (2019) or Mareček et al. (2020) focus on the use of multilingual-MT as a signal for learning language. As we will show, studying representations under different angles is required in order to highlight the differences underpinning distinct implementations of multilinguality.[1]

# 2 Experimental setup

## 2.1 Datasets

We focus on datasets derived from the OPUS-100 corpus (Zhang et al., 2020), built by randomly sampling from the OPUS parallel text collection (Tiedemann, 2012). We construct datasets containing 3, 6, 9, 12, 24, 36, 48, 60 and 72 languages other than English and refer to them as opus-03, opus-06, and so on. To test the impact on the model performance when adding languages, we build the datasets with an incremental approach, so that smaller datasets are systematically contained in the larger ones. Languages are selected so as to maximize the number of available datapoints—for training, zero-shot evaluation and probing—as well as linguistic diversity. See Appendix A for details.

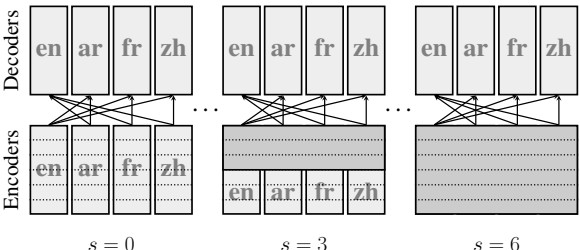

Figure 1: Example model architectures for varying number of shared encoder layers $s$. Modules with a light grey background are language-specific, modules with a dark grey background are fully shared.

## 2.2 Models

We train modular sequence-to-sequence Transformer models (Escolano et al., 2021), with 6 layers in the encoder and the decoder. Decoders are systematically language-specific, whereas encoders contain $s \in \{0, \ldots, 6\}$ fully-shared layers on top of $6 - s$ language-specific layers, as shown in Figure 1. We train distinct models for each value of $s$ and each dataset; due to the computational costs incurred, we consider $s \geq 2$ only in combination with datasets up to opus-12, as well as opus-36. Models vary along two axes: models trained on larger datasets are exposed to more languages, whereas models with higher values of $s$ share more parameters. When training models over a dataset, we consider the translation directions $L$-to-English, English-to-$L$, and a $L$-to-$L$ denoising task, for all languages $L$ in the dataset.[2] The noise model for the denoising auto-encoding objective follows Lewis et al. (2020). An illustration of opus-03 models is shown in Figure 1. Training details are given in Appendix B

# 3 Experiments

## 3.1 Task fitness: Machine Translation

The first aspect we consider is the models' performance on machine translation. We report BLEU scores in Figure 2. Where relevant, we also include supervised results for translation directions present in opus-06 so as to provide comparable scores.[3]

---

[1]Code available at: https://github.com/Helsinki-NLP/FoTraNMT/tree/who-would-win.

[2]I.e., a model trained over the opus-$n$ dataset is trained over $3n$ tasks: $2n$ translation tasks, plus $n$ denoising tasks for languages other than English.

[3]Note that all available zero-shot translation directions are systematically present in opus-06 and all larger datasets.

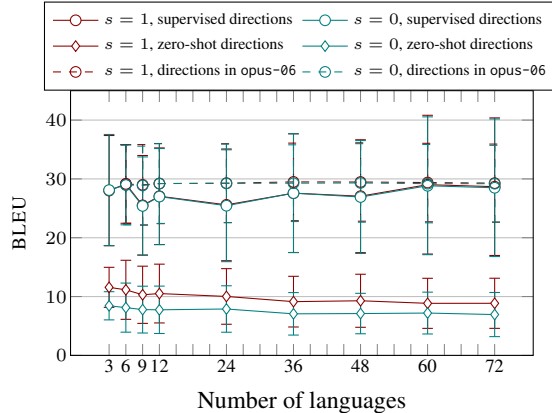

(a) Average BLEU scores per dataset size

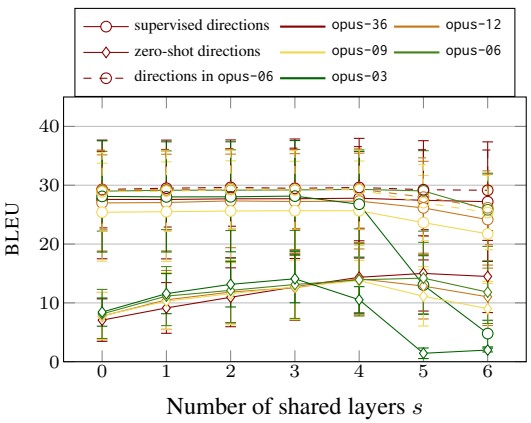

(b) Average BLEU scores per number of shared layers

Figure 2: Average BLEU scores

The most obvious trend present is that models trained on opus-03 with $s \geq 5$ underfit, and perform considerably worse than their $s < 5$ counterpart. Otherwise, models with an equivalent number of shared layers $s$ tend to perform very reliably across datasets: e.g., across all supervised translation directions we tested, we found that the maximum variation in BLEU scores for $s < 2$ was of $\pm 4.8$.[4] In Figure 2b, we also observe consistent improvement on zero-shot translation when increasing the number of shared layers $s$ from 0 to 4, and for opus-36 this trend only breaks when the full stack is shared ($s = 6$). Lastly, results in Figure 2a suggest that adding more translation directions decreases zero-shot translation performances, but this trend seems to reverse when a significant number of layers are shared ($s > 3$), as displayed in Figure 2b. In all, under the setup we consider here, it appears that *task fitness and zero-shot generalization are best achieved by sharing more parameters, rather than adding translation directions*—although ex-

cessive sharing also impacts performances.[5]

## 3.2 Language Independence: XNLI

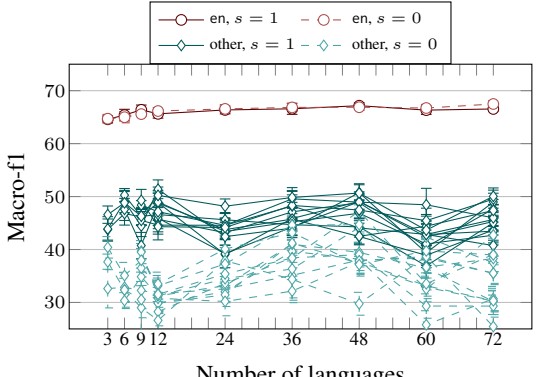

(a) Average XNLI scores per dataset size

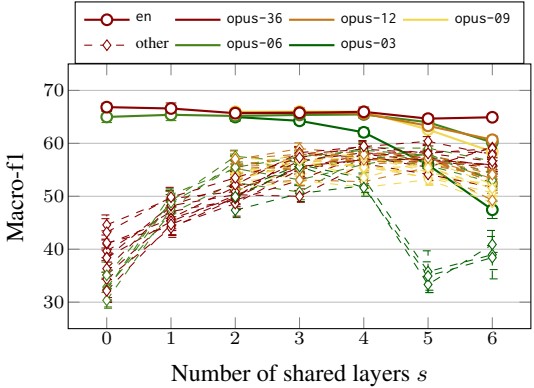

(b) Average XNLI scores per number of shared layers

Figure 3: Average XNLI macro-f1 scores

To test to what degree encoder representations are language-independent, we train classifier probes on XNLI (Conneau et al., 2018). We train models on English and report results for all languages: the gap between English and non-English performances quantifies how language-dependent the representations are. We report macro-f1 on the validation split; if no such split is available, we randomly select 10% instead. See Appendix C for details.

Figure 3 underscores that our English-centric scenario prevents language-independent encoder representations: English targets fare better than their counterparts. Variation seems driven by the number of shared parameters: in Figure 3a, models with $s = 1$ outperform models with $s = 0$, whereas in Figure 3b, higher values of $s$ tend to close the gap between English and other targets. Interestingly,

---

[4]See also Aharoni et al. (2019) or Conneau et al. (2020).

[5]Previous fully-shared models achieved high zero-shot performances, e.g. Johnson et al. (2017).

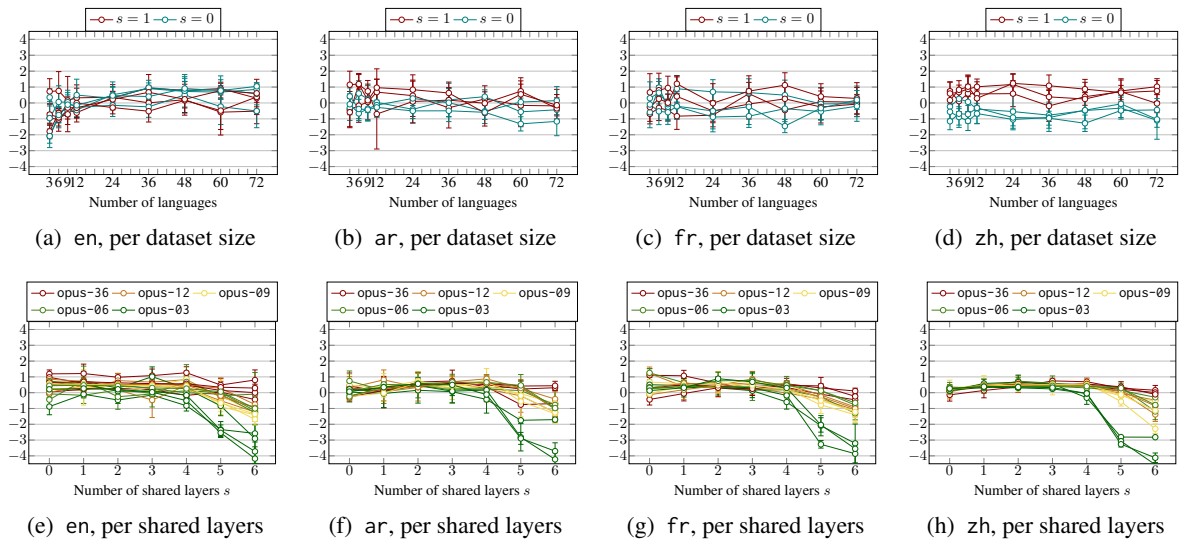

Figure 4: Average macro-f1 scores ($z$-scaled) on NLU monolingual tasks

higher values of $s$ yield lower f1 scores in smaller datasets, both for English and other languages. In particular, we observe a drop for all languages on `opus-03` with $s > 4$, matching the underfitting we saw in Section 3.1; this trend is also attested in all datasets except `opus-36`. But on the whole, *a greater number of shared parameters leads to more language-independent representations*.

### 3.3 Semantic Content: NLU benchmarks

To verify the semantic contents captured by our representations, we test them on monolingual GLUE-style benchmarks. We focus on benchmarks for languages present in `opus-03`: Arabic (ALUE, Seelawi et al. 2021), Chinese (CLUE, Xu et al. 2020), English (GLUE, Wang et al. 2018) and French (FLUE, Le et al. 2020). We select tasks that can be learned using a simple classifier; see Table 4 in Appendix C for a full list of the monolingual classification tasks considered. We follow the same methodology as in Section 3.2.

Results are displayed in Figure 4. Instead of plotting raw macro-f1 scores, we first $z$-normalize them so as to convert them to a comparable scale. Looking across datasets (Figures 4a to 4d), we do not see a clear variation; at best, we can argue English performances improves when using more language pairs. This is consistent with the English-centric scenario under which we trained our models. Arabic and Chinese results would suggest that $s = 1$ models fare better than $s = 0$ models, but this trend does not carry on convincingly for French.

Comparing across number of shared layers (Fig-

ures 4e to 4h) suggests this trend might be more complex: all languages tend to lose in accuracy for higher values of $s$, and this effect is all the more pronounced for non-English languages and models trained on smaller datasets. For instance, the optimal number of shared layers for Chinese is either $s = 3$ or $s = 4$, depending on the task under consideration and the number of language pairs in the training dataset, but the gain over $s < 3$ models is minimal. This differs crucially from what we observed in Section 3.1, where only $s = 6$ impacted BLEU scores, and in Section 3.2, where there was a clear improvement from low to mid values of $s$. In sum, probing encoder representations for their semantic contents paints a more nuanced picture, one where *semantic accuracy does not clearly align with task fitness or language-independence*.

## 4 Conclusions

We have studied whether different means of achieving multilinguality—sharing parameters and multiplying languages—bring about the same effects. What transpires from our experiments is that the two means are not equivalent: we generally observe higher performances and more reliable representations by setting the optimal number of shared parameters. Crucially, this optimum depends on the criteria chosen to evaluate representations: machine translation quality (Section 3.1), language independence (Section 3.2) and semantic accuracy (Section 3.3) all differed in that respect.

These two approaches are not dichotomous: it is possible to both scale the number of languages and

select optimal parameter sharing. What is possible may however not be practical. As guidance to NLP practitioners, we recommend spending effort on tuning the level of parameter sharing for the task at hand. Sharing either too little (0–1 layers in our experiments) or too much (sharing the entire encoder) results in sub-optimal performance overall, but the optimal number of layers to share depends on the task. Spending significant effort on acquiring data for additional language pairs may not yield improved representations past the initial stages of data collection (opus-03 in our experiments).

## Acknowledgements

This work is part of the FoTran project, funded by the European Research Council (ERC) under the EU's Horizon 2020 research and innovation program (agreement № 771113). We also thank the CSC-IT Center for Science Ltd., for computational resources and NVIDIA AI Technology Center (NVAITC) for the expertise in distributed training.

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

## A    Selected Languages

When constructing larger datasets, we select the additional languages based on four criteria:

(a) maximise the number of datapoints available for training

(b) the presence of zero-shot translation test sets

(c) the existence of XNLI data for the languages

(d) maximize language diversity in the dataset

The information we considered is listed in Table 1, with the exception of criterion (b): only languages in `opus-03` and `opus-06` are relevant to this criterion.

| ISO 2 | Dataset | Train size | XNLI |
|-------|---------|------------|------|
| ar | opus-03 | 1,000,000 | ✓ |
| fr | opus-03 | 1,000,000 | ✓ |
| zh | opus-03 | 1,000,000 | ✓ |
| de | opus-06 | 1,000,000 | ✓ |
| nl | opus-06 | 1,000,000 | ✓ |
| ru | opus-06 | 1,000,000 | ✓ |
| th | opus-09 | 1,000,000 | ✓ |
| tr | opus-09 | 1,000,000 | ✓ |
| vi | opus-09 | 1,000,000 | ✓ |
| bg | opus-12 | 1,000,000 | ✓ |
| el | opus-12 | 1,000,000 | ✓ |
| es | opus-12 | 1,000,000 | ✓ |
| bn | opus-24 | 1,000,000 | – |
| eu | opus-24 | 1,000,000 | – |
| fa | opus-24 | 1,000,000 | – |
| fi | opus-24 | 1,000,000 | – |
| he | opus-24 | 1,000,000 | – |
| id | opus-24 | 1,000,000 | – |
| it | opus-24 | 1,000,000 | – |
| ja | opus-24 | 1,000,000 | – |
| ko | opus-24 | 1,000,000 | – |
| lv | opus-24 | 1,000,000 | – |
| mk | opus-24 | 1,000,000 | – |
| sv | opus-24 | 1,000,000 | – |
| bs | opus-36 | 1,000,000 | – |

| ISO 2 | Dataset | Train size | XNLI |
|-------|---------|------------|------|
| cs | opus-36 | 1,000,000 | – |
| et | opus-36 | 1,000,000 | – |
| hu | opus-36 | 1,000,000 | – |
| is | opus-36 | 1,000,000 | – |
| lt | opus-36 | 1,000,000 | – |
| mt | opus-36 | 1,000,000 | – |
| ro | opus-36 | 1,000,000 | – |
| sk | opus-36 | 1,000,000 | – |
| sq | opus-36 | 1,000,000 | – |
| sr | opus-36 | 1,000,000 | – |
| uk | opus-36 | 1,000,000 | – |
| ca | opus-48 | 1,000,000 | – |
| da | opus-48 | 1,000,000 | – |
| hr | opus-48 | 1,000,000 | – |
| mg | opus-48 | 590,771 | – |
| ml | opus-48 | 822,746 | – |
| ms | opus-48 | 1,000,000 | – |
| no | opus-48 | 1,000,000 | – |
| pl | opus-48 | 1,000,000 | – |
| pt | opus-48 | 1,000,000 | – |
| si | opus-48 | 979,109 | – |
| sl | opus-48 | 1,000,000 | – |
| ur | opus-48 | 753,913 | – |
| af | opus-60 | 275,512 | – |
| cy | opus-60 | 289,521 | – |
| eo | opus-60 | 337,106 | – |
| ga | opus-60 | 289,524 | – |
| gl | opus-60 | 515,344 | – |
| gu | opus-60 | 318,306 | – |
| hi | opus-60 | 534,319 | – |
| ka | opus-60 | 377,306 | – |
| ne | opus-60 | 406,381 | – |
| nn | opus-60 | 486,055 | – |
| sh | opus-60 | 267,211 | – |
| xh | opus-60 | 439,671 | – |
| as | opus-72 | 138,479 | – |
| az | opus-72 | 262,089 | – |
| br | opus-72 | 153,447 | – |
| km | opus-72 | 111,483 | – |
| ku | opus-72 | 144,844 | – |
| nb | opus-72 | 142,906 | – |
| pa | opus-72 | 107,296 | – |
| rw | opus-72 | 173,823 | – |
| ta | opus-72 | 227,014 | – |
| tg | opus-72 | 193,882 | – |
| uz | opus-72 | 173,157 | – |
| wa | opus-72 | 104,496 | – |

| ISO 2 | Dataset | Train size | XNLI |
|-------|---------|------------|------|

Table 1: Languages selected matched with the first sub-dataset they appear in

## B Hyperparameters & Training details

**Hyperparameters**   Models were trained for a total of 100K steps to minimize the negative log-likelihood of the target translation. We accumulate gradients over all translation directions before back-propagation. We optimize our models using AdaFactor (Shazeer and Stern, 2018).

Training occurred on SLURM clusters of A100 NVIDIA GPUs. Each GPU contains the parameters for 3 languages (i.e., 9 translation directions); groups of 4 GPUs form a node. In other words, models for opus-03 were trained on a single A100 GPU, whereas models for opus-72 were trained over 24 A100 GPUs, distributed across 6 nodes. We did not go beyond opus-72 because this matches the largest setup in the computing cluster we used for our experiments. Using the modular training approach with unlimited compute resources, the ideal setup in terms of throughput would contain only one translation direction per GPU as it would allow concurrent training of all translation directions. However, this can introduce larger communication overheads unless all communication calls are not also performed asynchronously and concurrently. A detailed study assessing the training performance, including communication overheads, remains a subject for future work. In this study, all individual models were trained under 36 hours, cf. table 2.

|        | $s = 0$        | $s = 1$        |
|--------|----------------|----------------|
| opus03 | 1 day 08:15:00 | 1 day 09:46:00 |
| opus06 | 1 day 03:55:00 | 1 day 04:08:00 |
| opus09 | 1 day 04:04:00 | 1 day 03:44:00 |
| opus12 | 1 day 04:08:00 | 1 day 11:25:00 |
| opus24 | 1 day 04:44:00 | 1 day 04:37:00 |
| opus36 | 1 day 05:32:00 | 1 day 05:43:00 |
| opus48 | 1 day 05:46:00 | 1 day 06:30:00 |
| opus60 | 1 day 05:40:00 | 1 day 06:02:00 |
| opus72 | 1 day 05:53:00 | 1 day 06:21:00 |

Table 2: Models runtimes

Hyperparameters shared across all models are shown in Table 3; they were set *a priori* so as to

not use the validation split of opus-100, as it has been reported to significantly overlap with the test set (Yang et al., 2021). Input data is pre-tokenized using language-specific sentence piece models with 32,000 pieces, except for Chinese and Japanese, where we use 64,000 pieces.

| Parameter | Value |
|-----------|-------|
| src.seq. length | 200 |
| tgt.seq. length | 200 |
| subword type | sentencepiece |
| mask ratio | 0.2 |
| replace length | 1 |
| batch size | 4,096 |
| batch type | tokens |
| normalization | tokens |
| valid batch size | 4,096 |
| max generator batches | 2 |
| encoder type | transformer |
| decoder type | transformer |
| rnn size | 512 |
| word vec size | 512 |
| transformer ff | 2,048 |
| heads | 8 |
| dec layers | 6 |
| dropout | 0.1 |
| label smoothing | 0.1 |
| param init | 0.0 |
| param init glorot | true |
| position encoding | true |
| valid steps | 500,000 |
| warmup steps | 10,000 |
| report every | 50 |
| save checkpoint steps | 25,000 |
| keep checkpoint | 3 |
| accum count | 1 |
| optim | adafactor |
| decay method | none |
| learning rate | 3.0 |
| max grad norm | 0.0 |
| seed | 3435 |
| model type | text |

Table 3: Set of hyper-parameters shared across all our models

## C Classifiers training procedure

In Sections 3.2 and 3.3, we train classifier probes to investigate the information contained in the encoder spaces. All classifiers correspond to two-layer perceptrons with a hidden layer size of 128, dropout applied to the input layer, and trained with Adam

(Kingma and Ba, 2015) to optimize cross-entropy. We define sentence embeddings by simply taking the sum of the encoder output vectors; the input features of the classifiers are the concatenation of these sentence embeddings. For each set of targets, we train 10 classifiers with different random seeds and report the mean and standard deviation of macro-f1 scores. In Section 3.2, we set the learning rate for XNLI to $5 \cdot 10^{-5}$ with a dropout of $p = 0.1$ and use minibatches of 100 examples. Note that we consider each language in XNLI as a different set of targets, and therefore use different classifiers to compute macro-f1 scores.

| | Dataset | Task | Size |
|---|---|---|---|
| ar | NSURL-2019 Task 8 | question similarity | 10,797 |
| | OSACT4 Task-A | offensive speech detection | 6,839 |
| | OSACT4 Task-B | hate speech detection | 6,839 |
| en | COLA | linguistic acceptability | 8,551 |
| | MRPC | sentence similarity | 3,668 |
| | QNLI | NLI | 104,743 |
| | QQP | question similarity | 363,846 |
| fr | PAWSX | paraphrase detection | 49,399 |
| | STSB | paraphrase detection | 5,749 |
| | XNLI | NLI | 392,702 |
| zh | AFQMC | question similarity | 34,334 |
| | CMNLI | NLI | 391,783 |
| | TNEWS | news topic classification | 53,360 |

Table 4: NLU monolingual classification tasks

The classification tasks selected for studying the semantic contents of encoder representations in Section 3.3 are shown in Table 4. Due to the limited number of usable tasks in FLUE, we also include a STSB French translation[6] which we binarize by considering similarity judgments $> 3$ as indicating near-paraphrases. Classifiers discussed in Section 3.3 are trained for 10 epochs with a dropout of 0.1 and a learning rate of $5 \cdot 10^{-5}$, using minibatches of 100 datapoints. We reduced the number of epochs to 5 for all Arabic tasks and used minibatches of 10 examples for the OSACT4 shared tasks A & B due to the longer length of the training examples.

# D  Limitations

## D.1  Material Limitations

As stated in the introduction, we make multiple explicit assumptions that limit the scope of this research. It is plausible that parameter-sharing in the decoder or that replicating our experiments in a non-English-centric scenario will yield a different set of conclusions.

Also worth highlighting are the computational requirements underlying this work: the most demanding experiments require up to 24 A100 NVIDIA GPUs. A side-effect of these demanding computational requirements is that we have not been able to replicate model training across multiple seeds, and therefore report results based on a single model per dataset and number of shared layers. It is also plausible that greatly scaling up the total number of parameters in the networks would affect the conclusions.

Lastly, our use of classifiers to probe for language independence and semantic contents of the representations can be discussed. We have avoided discussing the raw performances of our classifiers, and instead discussed the trends that we observed across our different MT models. Results from our classifiers should be taken as indicators of the aspects we are trying to probe, rather than accurate measures of said aspects: replication studies and further evidence from other settings would be required to establish our models' performances on the criteria we outlined.

## D.2  Ethics Considerations

In the present paper, we have argued against adding languages if practical implementation costs are a relevant constraint. We acknowledge that this recommendation may push NLP researchers and engineers towards constructing models specifically for high-resource languages, which would further the coverage gap between low- and high-resource languages.

Nonetheless, it must be stressed that our experiments say nothing of linguistic diversity, as we have ensured that even our smallest dataset (opus-03) would contain maximally different languages. Also relevant to the discussion at hand is that one scenario where practical implementation costs are a known constraint is that of developing low-resource languages systems and NLP tools. We believe that providing evidence as to which approach is most effective can prove valuable in such scenarios as well, so as to ensure that efforts can be focused on the most viable path towards endowing lower-resource languages with more efficient and suitable tools.

---

[6]https://huggingface.co/datasets/stsb_multi_mt