# OpenReview forum: "Dozens of Translation Directions or Millions of Shared Parameters? Comparing Two Types of Multilinguality in Modular Machine Translation"
_NoDaLiDa/2023/Conference — NoDaLiDa 2023_

### Official Review · Reviewer_rtje · 2023-03-09
**A systematic study of variants of modular machine translation with much data and less discussion**

**Rating:** 6
**Confidence:** 2

**Review:**

The paper picks up on the idea of modular machine translation (Escolano et. al 2021) and report results on varying two different variables of that approach, the number of shared layers and the number of languages. The work is English-centric meaning that only translation directions in and out of English are considered. Another limitation is that layers are shared only in the encoder part. Still, there is a lot of module training that has been performed and a lot of test data being reported. The results are summarized as a kind of advice to practitioners: spend efforts on tuning the level of parameter sharing for the task at hand. In more detail, the paper indicates that parameter sharing is beneficial only up to a point (3-4 shared layers out of the 6 used for the models), while adding many more languages in this English-centric fashion has varying and often negative effects.

The experiments are performed systematically, but as a number of variables are fixed only covers a small part of the design space (which is admitted). The presentation is terse and not always easy to follow unless you've done the same type of experiments yourself. The work actually deals with layer sharing, while the title talks of 'millions of parameters'. I looked for information on how many parameters the models actually use, but didn't find one; Table 2 lists hyper-parameters shared across models, but no explicit figure matching the parameters mentioned in the title.

One aspect of multilinguality that the paper considers is language independence. This is measured by differences in results on XNLI data for English as compared to results for the other languages and demonstrates a large gap, which is somewhat closed when more layers are shared. It would have been interesting also to see figures on translations into vs. out of English, rather than just BLEU averages for all translations as in Figure 2.

I judge the quality of the work as fair, but the clarity of the presentation somewhat lacking. It is also hard to judge the novelty of the approach and the results. Although there are many references to similar work, the paper does not report any hypotheses as to what results could be expected nor compares the results with other work.

**Paper Type:**

Long paper

---

### Official Review · Reviewer_4PpF · 2023-03-12
**Dozens of Translation Directions or Millions of Shared Parameters? Comparing Two Types of Multilinguality in Modular Machine Translation 003 054**

**Rating:** 8
**Confidence:** 5

**Review:**

Negative comments:
1. The paper should include citations for modular NMT, such as the one found in https://aclanthology.org/2020.emnlp-main.476.pdf.
2. I don't understand why 24 GPU-s are needed.
3. The benefit of using the classifier is not clearly explained in the paper.
4. Some of the figures (e.g., Figures 2.b, 3.b, and 4) are hard to read and could be improved for better clarity.

Positive comments:
1. The table in Appendix 2 that covers all of the hyper-parameters is a useful addition.
2. The experiments with multiple languages are valuable and provide important insights.
3. The graphs and figures contain a lot of interesting information, although some may be overwhelming.

Improvements:
One possible improvement for the paper is to include a table or graph showing the number of parameters and training times (GPU-hours) for each of the models tested in the experiments. This information could provide readers with a clearer understanding of the computational requirements of the proposed method and how it compares to other approaches.

General comments:
This is a well-written paper that provides valuable research on multilingual machine translation. However, the research is limited to an English-centric scenario, which may limit its applicability in production settings where multiple language pairs are used. It would be interesting to explore the performance of the proposed method in a more diverse multilingual setting. Overall, the paper is a valuable contribution to the field of natural language processing and machine translation.

**Paper Type:**

Short paper

---

### Official Review · Reviewer_ELqS · 2023-03-13
**Dozens of Translation Directions or Millions of Shared Parameters? Comparing Two Types of Multilinguality in Modular Machine Translation**

**Rating:** 7
**Confidence:** 3

**Review:**

This work explores the encoder representations with multilingual modular NMT models using a different number of languages and shared encoder layers. The work is nicely structured and correctly formatted. The scope of the article is clearly defined, but I found it a bit difficult to understand the goal from the abstract and introduction as the authors jump straight into their topic without easing in the reader first, however, it became easier to follow once I read the experiment description.

The experiments are done systematically on a large scale and the insight is fascinating, however, the decision of using English-centric training may cause some issues when so much of the evaluation focuses on zero-shot performance. Although the authors get some zero-shot performance (probably thanks to the denoising task), it is uncertain whether the same conclusions would be drawn if the selection of language pairs was less restrictive and more realistic. I would also appreciate additional information on the denoising task.

**Paper Type:**

Short paper

---

### Decision · Program_Chairs · 2023-03-17

Accept